# Can (We Make) *Bacillus thuringiensis* Crystallize More Than Its Toxins?

**DOI:** 10.3390/toxins13070441

**Published:** 2021-06-26

**Authors:** Guillaume Tetreau, Elena A. Andreeva, Anne-Sophie Banneville, Elke De Zitter, Jacques-Philippe Colletier

**Affiliations:** Univ. Grenoble Alpes, CNRS, CEA, Institut de Biologie Structurale, F-38000 Grenoble, France; guillaume.tetreau@gmail.com (G.T.); Elena.Andreeva@ibs.fr (E.A.A.); Anne-Sophie.Banneville@ibs.fr (A.-S.B.); Elke.De-Zitter@ibs.fr (E.D.Z.)

**Keywords:** pore-forming toxin (PFT), pesticidal protein, bacteria, crystals, crystalline formulation, bioinsecticide, biotechnology, structural biology

## Abstract

The development of finely tuned and reliable crystallization processes to obtain crystalline formulations of proteins has received growing interest from different scientific fields, including toxinology and structural biology, as well as from industry, notably for biotechnological and medical applications. As a natural crystal-making bacterium, *Bacillus thuringiensis* (*Bt*) has evolved through millions of years to produce hundreds of highly structurally diverse pesticidal proteins as micrometer-sized crystals. The long-term stability of *Bt* protein crystals in aqueous environments and their specific and controlled dissolution are characteristics that are particularly sought after. In this article, we explore whether the crystallization machinery of *Bt* can be hijacked as a means to produce (micro)crystalline formulations of proteins for three different applications: (i) to develop new bioinsecticidal formulations based on rationally improved crystalline toxins, (ii) to functionalize crystals with specific characteristics for biotechnological and medical applications, and (iii) to produce microcrystals of custom proteins for structural biology. By developing the needs of these different fields to figure out if and how *Bt* could meet each specific requirement, we discuss the already published and/or patented attempts and provide guidelines for future investigations in some underexplored yet promising domains.

## 1. Introduction

A crystal is a regular tridimensional arrangement of identical molecules or complexes of molecules. As opposed to solutions (liquids) and aggregates (amorphous solids), where molecules are randomly distributed and oriented, a crystal is characterized by a symmetry with a limited number of unique molecule orientations imposed by the crystal lattice [1]. This is associated with a number of properties that are exploited by different scientific fields. For example, the high level of crystal symmetry allows diffracted X-rays to be detected and analyzed to solve the structures of biological macromolecules, a key knowledge to finely understand their function(s). Macromolecular X-ray crystallography (MX) is accordingly the most prolific method in structural biology, accounting for more than 90% of the structures deposited in the Protein Data Bank (PDB) [2]. Additionnally, crystals represent a means to provide molecules at high concentration and with properties that can be tailored to provide long-term storage, controlled release, and retained activity. This is of particular interest for drug development and delivery, notably of pharmaceuticals [3,4], but also for catalysts that can be formulated for large-scale industrial applications [5,6]. Crystals therefore hold the promise of multiple applications, from the most fundamental academic research purposes to the development of innovative biotechnological products [6]. However, the crystallization of macromolecules, and notably proteins, is a process hardly predictable due to the many parameters affecting the nucleation and growth of crystals [7,8,9]. Crystallization of proteins implies that they are intrinsically capable of sufficiently strong crystal packing interactions to retain order in the long range. Not all proteins are able to form such interactions, and even if they do, the conditions to obtain a crystal are generally highly specific and involve many different parameters. In vitro, crystal formation depends on the purity and concentration of the protein, the nature and concentration of the protein precipitant, the nature of the buffer, the pH, the temperature, etc. [10]. To explore and identify the conditions of crystallization for one particular protein, a classical approach consists of establishing its crystallization phase diagram (Figure 1A) [11]. Different methods of crystallization have been developed in vitro to generate different crystallization trajectories to either favor single macrocrystals or multiple microcrystals [10]. Despite the development of several different crystallization procedures during recent decades [12], the identification and optimization of crystallization conditions leading to a desired set of crystal properties can sometimes be a long and tedious empirical exploration that may be paved paved with failure [4,5,13].

Protein crystallization is ubiquitous in nature [24]. In vivo crystallization has been known for decades and is notably associated with diseases [25,26], encapsulation [27,28], and storage of nutritive proteins [29,30] and virulence factors [22,31]. In humans, Charcot–Leyden crystals, which are composed of the eosinophil galectin-10 protein, are characteristics of allergy-induced asthma [32]. In viviparous cockroaches, mothers feed their offspring with “milk” constituted of crystals of heterogeneous glycosylated proteins [33]. In Baculoviruses, polyhedrin proteins are produced at high concentration during the late stage of insect cell infection to protect the virion by encapsulating it into an intracellular crystal called polyhedral [34]. Using living organisms to produce proteins in the form of nano/micrometer-sized crystals directly in cells or cell compartments could be of high interest, notably as it bypasses the need to extract, purify, and crystallize proteins in nonphysiological conditions [35]. The in vivo crystallization of recombinant proteins has already been attempted and observed in plant [36,37], animal [38,39,40], insect [41,42,43], and bacterial [44,45] cells. Among all these living organisms able to produce crystals, one is of particular interest, the bacterium *Bacillus thuringiensis* (*Bt*).

*Bt* has evolved over millions of years into a natural crystal maker, with hundreds of subspecies each crystallizing one or more toxins. These toxins exhibit drastically different tridimensional structural organization [46] and recognize contrasting protein receptors within the gut of insects from various invertebrate phylogenetic groups. Despite these differences driven by their contrasting modes of action, the pathways of crystallization of each of these toxins lead to crystals with shared characteristics, namely highly intrinsic organization of toxins within the crystal, long-term stability in aqueous environment, and specific dissolution of crystals to alkaline medium. Using *Bt* to produce custom nanocrystals in vivo could therefore be envisioned by capitalizing on the knowledge acquired during the last decades on the mode of action of *Bt* toxins and their mechanisms of crystallization. A variety of tools allowing the genetic manipulation of *Bt*, notably including a large set of shuttle vectors (Figure 1B), have been developed to modify *Bt* strains and stably express a large variety of toxins and toxin complexes [14,47]. In addition, several acrystalliferous *Bt* strains, i.e., curated from their plasmids that carry toxin genes, have been developed for the recombinant production of toxin crystals [48,49,50,51].

In the present article, we selected three fields for which the combination of the knowledge acquired and the molecular biology tools developed have benefited or could benefit the recombinant production of custom proteins as crystals in *Bt*. This selection is obviously subjective and is organized from the most studied and referenced domain to the most underexplored, albeit promising, one. *Bt* crystallization system could be hijacked to develop new bioinsecticidal formulations based on rationally improved crystalline toxins (Section 2), to functionalize crystals with specific characteristics for biotechnological and medical applications (Section 3), and to produce microcrystals of custom proteins for structural biology (Section 4). Each of these three aspects is discussed with regard to the existing literature, if any, and guidelines are provided for promoting further developments in the promising aspects of each field.

## 2. Producing New Crystalline Toxins for the Development of Innovative Bioinsecticides

*Bt* has been increasingly used over the last century for an environmentally friendly integrated pest control strategy [52]. Commercialized biopesticides for topical applications have essentially relied on few approved WT strains with narrow host spectra, namely *Bt* subsp. *kurstaki* (*Btk*) and *aizawai* (*Bta*) against Lepidoptera, *Bt* subsp. *israelensis* (*Bti*) against Diptera, and *Bt* subsp. *tenebrionis* (*Btt*) against Coleoptera [53,54]. Limited host spectrum is one of the major advantages of *Bt*-based bioinsecticides, which allows off-target effects to be limited while reducing the range of pest insects that can be controlled with one given *Bt* subspecies. Different strategies have been published [22,54], patented [55,56], and commercialized [53,54] to bypass these limitations. The rotational or pivotal use of a given *Bt* subspecies with another *Bt* subspecies (or another entomopathogen organism) allows taking advantage of the properties of each toxin mixture. For example, mosquito control can be achieved by combining the *Bti*’s four-toxin mixture with crystals of the binary toxin Tpp1Aa/2Aa (formerly BinAB) produced by the bacterium *Lysinibacillus sphaericus* (*Ls*). When used together, they reduce the risk of resistance development in mosquito populations and increase the range of mosquito species targeted [57,58,59]. Further research has been devoted to engineer *Bt* strains, either by conjugation or recombination, with the aim of extending the set of toxins they produce. Tpp1Aa/2Aa toxins were notably introduced in *Bti* to gain the advantage of the extended toxin mixture while reducing the need for multiple crystal production processes [60]. Similarly, Cry11Ba from *Bt* subsp. *jegathesan* (*Btj*), a homolog of *Bti*’s Cry11Aa that exhibit higher toxicity and an extended host spectrum [61], was produced in *Bti*. By using acrystalliferous strains, such as *Bti* 4Q7 [48], novel toxin combinations can be created, for example, concomitantly producing the toxins Cyt1Aa from *Bti*, Cry11Ba from *Btj*, and Tpp1Aa/2Aa from *Ls* [62]. A similar approach was used to generate and commercialize *Bt* strains with a large pest spectrum activity by appending anticoleopteran toxins, such as Cry3Aa from *Btt* or Cry3Bd from *Bt* subsp. *kumamotoensis*, to antilepidopteran toxins produced by *Btk*, yielding to the Foil^®^ and Raven^®^ products, respectively [53].

Tailoring of the toxicity level and host range of *Bt* strains can be achieved not only by the discovery of new toxins and by creating *Bt* strains with new toxin combination but also by directly engineering the toxins themselves. The characterization of some aspects of the mode of action of toxins has driven the modification of toxins, in particular to bypass any resistance mechanism developed by insects, especially target site modifications [63,64]. For example, Cry1AMod toxins were constructed from either Cry1Ab or Cry1Ac by removing the helix α-1 and part of helix α-2 from the N-terminal part of proteins [65] based on the observation that Cry1A binding to a cadherin receptor leads to the cleavage of these residues and increases toxicity [66]. For Cry3Aa, this is the incorporation of chymotrypsin cleavage sites in a loop between the helices α-3 and α-4 of domain I, which yields the toxin mCry3Aa, thereby accelerating its activation and leading to an increased toxicity and widened host spectrum [67]. Phylogenetic analyses of the domains from three-domain Cry toxins revealed that part of the toxin evolution leading to the diversity of *Bt* toxins relied on homologous recombination of their domain III, known to be involved in recognition and binding to receptors and in toxin oligomerization [68]. This led to many interchanges between domains III, or part of it, to tailor the toxin toxicity and spectrum. Domain III of Cry3Aa was replaced by that of Cry1Ab, leading to eCry3.1Ab with improved toxicity to *Diabrotica virgifera* by targeting different gut receptor(s) [69]. The replacement of domain III of Cry1Ab by that of Cry1F (yielding the new toxin named Cry1A.105 [70]; Figure 2A) or Cry1C, both from *Bta*, led to a widened host spectrum [71] and increased toxicity against *Spodoptera exigua* [72], respectively. Similar results against *Spodoptera frugiperda* were obtained by combining domains I and II of Cry1Ba, domain III of Cry1Ca, and the “crystallization domain” of Cry1Ac [73]. Of note, several of these recombinant/modified toxins have been introduced in genetically modified (GM) plants directly producing the toxins, thereby circumventing several limitations of topical insecticide applications (e.g., limited persistence, difficult timing of applications, and higher production costs) [53]. Cyt1Aa toxin from *Bti* has also been used to generate chimeras, expanding its host spectrum to other mosquito species when fused to the binary Tpp1Aa/2Aa (formerly BinAB) from *Ls* [74] and to lepidopteran when the recognition loop 3 of domain II from Cry1Ab was inserted between its loops 1 and 2 [75]. Interestingly, interdomains exchanges conducted between Cry11Aa and Cry11Ba from *Bti* and *Btj*, respectively, highlighted that while some domain combinations improved the mosquitocidal activity, others completely abolished the ability of the chimeras to form inclusions [76,77]. This exemplifies how different domain shuffling using two “sister” toxins, which should crystallize through a similar, albeit thus far uncharacterized, pathway and exhibit alike mode of actions, can alter the formation and stability of their crystal, along with their specificity and toxicity. This provides key information to guide modifications of the properties of both the toxin and its crystal.

Of note, fusion of different toxins or toxin parts with non-*Bt* toxic proteins has been performed to combine their insecticidal properties for pest control. For example, Cry1Ac fusion with Av3 toxin from the Cnidaria *Anemonia viridis* [79], WTX-XI toxin from the spider *Ornithoctonus huwena* [80], ω-ACTX-Hv1 toxin from the spider *Hadronyche versuta* [81], or chitinases of various origins [82,83] all led to increased toxicity toward lepidopteran species. Although these chimeras generally generated inclusions when produced in *Bt*, their crystallinity and overall stability is yet to be demonstrated in order to develop new, reliable bioinsecticidal products. Their ability to form crystals rather than mere inclusion bodies may depend on the toxin but also on the fused protein, and not all combinations may lead to crystalline inclusions. Further studies combining different toxins and cargo proteins with different characteristics (e.g., molecular weight, isoelectric point, hydrophobicity, etc.) should be conducted and complemented by crystallinity investigations using suitable X-ray sources.

Strong phenotypic changes on toxins can also be obtained by a “surgical strike” strategy based on the modification of only few selected amino acids of high importance. Based on this approach, some engineered toxins have been developed, patented, and commercialized, such as Cry1Da_7, a Cry1Da triple mutant (S282V-Y316S-I368P) with >50-fold increased toxicity toward the lepidopteran *Helicoverpa zea* [73], and Cry51Aa2.834_16, a modified Mpp51Aa (formerly Cry51Aa) toxin containing eight point mutations and three-residue deletion leading to a >200-fold increased toxicity to the hemipteran *Lygus sp.* [78] (Figure 2B). Hundred- to thousand fold increases in mosquitocidal activity could also be obtained by four point mutations in Cry4Ba [84] and by four point mutations combined with a five-residue deletion in Cry19Aa [85]. Structural studies, notably based on X-ray crystallography and modeling, greatly help in identifying key residues to be mutated [86,87]. Moreover, structures solved directly from crystals grown in vivo provide insights into the crystallization pathways to design mutations affecting not only toxin activity and host spectrum but also crystal formation and stability. We recently used the in vivo structure of Cyt1Aa protoxin to show that the size, shape, production yield, pH sensitivity, and toxicity of Cyt1Aa crystals grown in *Bti* could be controlled by single amino acid substitutions [88]. Additional crystallographic studies on crystals grown in vivo are therefore expected to further extend and rationalize the strategy of toxin improvement through point mutations.

## 3. Functionalizing Toxin Crystals for the Development of New Biotechnological Tools

Protein crystals have gained interest in a variety of research domains as their properties, especially their high intrinsic order and porosity, could allow stabilizing functional proteins within the protein crystal scaffold for application in biotechnology and medicine [6]. The possibility of using protein crystals as a polyvalent nanomaterial able to concentrate, stabilize, and protect functional proteins has been envisaged for a large range of applications, notably including biosensing (i.e., detection of relevant molecules by the use of biological macromolecules), biotemplating (i.e., assembly of inorganic nanostructures guided by the protein crystal scaffold), catalysis, and vaccine/drug delivery [3,4,5,6]. Extensive engineering efforts have been made by acting on the crystal packing interfaces, notably through cross-linking [89,90], to tailor the crystal properties and create new scaffolds leading to potentially new crystal functionalities but also by finely tuning the pore structure and associated physicochemical characteristics to adjust the properties of the cargo protein [6].

Some *Bt* toxins offer a stable crystalline framework perfected over millions of years of evolution. Using *Bt* toxins crystals to entrap cargo proteins for either of the applications mentioned above could therefore facilitate the production of functional crystalline formulations by direct production in vivo, thereby bypassing the need for laborious protein extraction, purification, and in vitro crystallization. Michael K. Chan’s group has provided the most comprehensive investigation to date on this aspect. They published [45,91,92,93,94,95] and patented [96,97,98] a strategy involving different Cry toxins and more specifically Cry3Aa, which they selected for further investigations for several reasons. Cry3Aa is a toxin with high self-assembly propensity that is able to crystallize both in vitro and in vivo with similar crystal packing interfaces, indicating that its crystallization process is strongly driven by the toxin itself, pending an adequate toxin concentration [31,99,100]. Moreover, the checkerboard-like crystal packing of Cry3Aa offers large solvent channels that could be able to accommodate a cargo protein up to ~50 Å in diameter (Figure 3) [31,45]. This contrasts with other toxins, such as Cyt1Aa, which pack into crystals that might be too dense to allow a cargo protein to cocrystallize [88], although recent attempts suggest that it might not be an insuperable barrier [74].

Enzymes are increasingly used for industrial applications for biocatalytic processes of high economic importance, in particular for food processing, biofuel production, and natural gas conversion, and strategies to immobilize and boost the efficacy of these catalysts are particularly sought after [5,101,102]. For these reasons, research has been performed to provide a proof-of-concept of their approach using lipases from different origins with the aim of developing nanocrystalline formulations of a biocatalyst for the production of biodiesels [45,92,93]. It was shown that crystals of Cry3Aa allowed proteins up to 32 kDa to freely diffuse in the crystal after in vitro coincubation [95]. However, the cargo protein tended to be released from the crystal upon washing with buffer [93,95], suggesting that the protein might not succeed in being deeply buried into the solvent channels of the toxin crystals. This problem was circumvented by performing in vivo entrapment of the cargo protein, either by fusing the protein to the toxin [45,92,93] or by simply coexpressing them during the sporulation of the bacterium [93]. Both strategies revealed that the crystals successfully incorporated the cargo protein at high concentration and that the latter retained its catalytic activity. Moreover, crystals allowed the enzymes to be protected from degradation and to sustain their catalytic activity over several successive cycles of oil-into-biodiesel conversion [45,92,93]. They also showed that Cry3Aa toxin could be modified to favor the addressing of the cargo protein to the solvent channel by deleting the last 19 C-terminal amino acids before fusing with the lipase, which led to improved catalysis [45], although the yield remained lower than when the lipase was entrapped by simple coexpression with the toxin during *Bt* sporulation [93]. Altogether, these results highlight the versatility of the Cry3Aa crystallization system to entrap functional catalysts of high economic importance and the possibility to tailor the system for improving the efficacy of both cargo protein entrapment and catalytic activity. Although it is unclear whether all the inclusions that were obtained possessed the intrinsic organization of a crystal, it was shown that they managed to retain a sufficiently high quantity of functional protein for a phenotype to be observed and quantified [93]. This will hopefully fuel further research investigating additional *Bt* toxin/cargo protein combinations to explore the full potential and establish the limits of this system.

The same group also explored the potential of Cry3Aa crystals for drug delivery [91,94,95]. The use of crystalline formulations for the storage and delivery of biopharmaceutical proteins has received increasing interest during the last decade [13,103,104] as it offers many advantages associated with their higher stability, long-lasting and progressive diffusion within the organism, and long-term storage [3,4]. The major limitations to the widespread use of crystalline formulations is the lack of expertise on large-scale crystallization, the high heterogeneity of crystalline productions, and the absence of simple and straightforward crystal purification methods [3]. In vitro cocrystallization processes have been envisaged, notably for efficient drug delivery, but their development is hampered by similar practical limitations [13,105]. This explains why the number of crystalline formulations produced and commercialized is still limited [106]. Insulin was the first, and for long time the only, therapeutic protein produced and commercialized in a crystalline formulation [3]. In vivo crystallization, notably using *Bt*, could allow an expansion of the set of crystalline biopharmaceuticals in the market. Cry3Aa was shown to successfully entrap antimicrobial peptides and myoglobin, which could be further taken up by macrophages and cancer cells, respectively [94,95]. Moreover, by using only the first domain of Cry3Aa, an important domain for Cry3Aa crystallization in *Bt* [31,107], for fusion with myoglobin, results comparable to those with fusions using the full toxin were obtained, indicating that only some parts of the toxin might be sufficient to stabilize the cargo protein. However, the crystallinity of such inclusion has not been verified, and further experiments are needed to explore its effect on crystal-packing interfaces and inclusion properties.

When it comes to biopharmaceuticals, the crystal carrying the cargo protein must be biocompatible, i.e., innocuous for the receiving organism. Two additional precautions must therefore be taken into account, namely the need for innocuous cocrystallizing toxin and the possibility to get rid of 100% of the remaining living organisms [6]. For the former, although *Bt* Cry toxins require specific receptors that are absent from mammals [52] and should therefore be safe for therapeutic applications, they could be engineered by strategic point mutations known to abolish their toxicity while not impeding their crystallinity. This has already been demonstrated for toxins like Cyt1Aa (e.g., mutants V122E [108], K154A [109], or Q168E [88]), Cry1Ac (e.g., A92D, N166D, or L167F [110]), Cry3Aa (e.g., V155F-S156M-S157R [111]), Cry11Aa (e.g., R90E [112], S259A [113], or V262A [114]), and Cry11Ba (e.g., G257A or I263A [115]).

## 4. Microcrystallization Platform for Structural Biology

Protein crystals were discovered by chance more than 150 years ago and initially used for protein purification, but their main contribution to science is concomitant with the advent of X-ray crystallography in the late 1930s for the determination of three-dimensional structures of biological macromolecules [116,117]. The combination of the development of multiple crystallization methods [9,10,12], the implementation of cryogenic solutions limiting the propagation of X-ray-induced radiation damage [118], and the easy access to finely tuned and reliable X-ray synchrotron sources has allowed over 130,000 protein structures to be solved over the last 20 years [2]. It is therefore a method of choice for the determination of protein structures at high resolution. In synchrotron facilities, diffraction data are generally collected from a single oscillating macrocrystal (10–100 µm) to collect a complete dataset for reconstructing structures. Unfortunately, fragile macromolecules of complex architecture, such as membrane proteins or large protein complexes, and most proteins crystallized by living organisms generally form nano- to micrometer-sized crystals that are not amenable to structure determination by conventional oscillation methods at synchrotron sources. This is notably due to the progression of X-ray-induced radiation damage that is only delayed and not eliminated by flash-cooling (i.e., cryogenic conditions where the crystal is quickly frozen and maintained at a temperature of ~100 K). In practice, this means that the smaller the crystals, the more radiation sensitive they are and the more crystals are required to obtain a radiation-damage-free dataset [119,120]. The development of serial crystallography and the advent of X-ray free-electron lasers (XFEL) allowed this limitation to be bypassed. XFELs deliver the same amount of photons as synchrotrons but within tens of femtoseconds versus few milliseconds for the latter, allowing the collection of high-resolution diffraction patterns at room temperature (RT) before radiation damage onset via the so-called “diffraction before destruction” approach [121]. The most immediate consequence is that each crystal only contributes a unique diffraction pattern before it is destroyed by the X-ray beam, so thousands of homogenous microcrystals are required to be injected serially to collect a sufficient number of indexed patterns from different microcrystals in different orientations to determine the protein structure (Figure 4) [122,123].

Several in vitro crystallization procedures have been developed to produce the high quantity of diffraction-grade microcrystals required for such an approach [10]. However, establishing reliable and reproducible procedures remains a laborious process that is very demanding in term of manpower and time and has an unpredictable outcome [7,8,9,12]. In vivo crystallization has recently emerged as a sound alternative to produce the crystalline samples required for such experiments [24,35,124]. In addition to facilitating crystal production and purification, it would also permit the characterization of crystalline proteins as they naturally occur in the cellular context, i.e., protected from oxidative stress and sometimes in the presence of propeptides before protein maturation [31,44,88,125], and/or in complex with natural ligands subselected from the cytosolic pool of substrate [40,126]. The use of animal and insect cells has been proposed for the further development of in vivo crystallization [42,127], but major limitations hinder its establishment as a reliable platform for the production and isolation of pure nano/microcrystals. These include (i) cell death upon crystallization, thus raising the question of whether the crystalline protein could suffer oxidative stress and if complexed ligands are selected from the large pool of cytosolic metabolites present in a healthy cell or from the leftovers found in a dying cell [125]; (ii) the production of generally fragile needle-like crystals that are difficult to extract and are unstable outside the cell [40,124,125]; and (iii) a serendipitous, unnatural pathway of crystallization, offering little command on crystal size and diffraction quality, thereby limiting the possibility to learn how to engineer self-crystallizing interfaces in vivo from observations made at crystal packing interfaces. In contrast, *Bt* is a natural nanocrystal maker for a large array of proteins with contrasting size and structures, which it can crystallize through finely regulated crystallization pathways [46,128,129]. Moreover, it has been shown that high-resolution structures of toxins recombinantly expressed in an acrystalliferous *Bt* strain could be solved from diffraction datasets collected at XFELs directly from crystal-containing *Bt* cells [31] as well as from purified crystals [31,44,88]. The knowledge acquired and the molecular tools developed for *Bt* combined with the multiple successes in using *Bt* toxins as crystallization vessels for cargo proteins (see Section 2 and Section 3) point to *Bt* as a model of choice for the in vivo crystallization of recombinant proteins. Of note, the absence of electron density from synchrotron data obtained using Cry3Aa crystals that incorporated a lipase during their growth in vitro [93] suggests that extensive efforts must be done to stabilize the interactions between the crystalline toxin and the cargo protein within the solvent channel. This will require fine-tuning of the sequence and characteristics of the toxin and/or cargo protein to stabilize the protein within the solvent channel while preventing the motions within the cargo protein from becoming restrained should one want to elucidate the dynamic aspects of its function [130,131]. Exploring new microcrystallization methods is of utmost interest considering the recent development of new solutions for structural biology. The increasing number of beamlines dedicated to serial injection at synchrotron facilities and the progressive upgrade of these X-ray sources, which has increased their brilliance by approximately two orders of magnitude to make them so-called extremely brilliant sources (EBS) [132], are opening new opportunities in structural biology. Altogether, they hold the promise to significantly increase the set of biological systems to be investigated by X-ray crystallography and electron microscopy and to enable access to structural information to a wider range of scientists from various disciplinary fields.

## 5. Conclusions

The ability of natural *Bt* strains to make crystals of toxins has been largely exploited for the development of biological insecticide formulations for an integrated pest control strategy. In this article, we covered various successful and failed attempts aimed at hijacking the crystallization machinery of *Bt* for it to be used as a custom crystal biofactory. We saw that bioinsecticide improvement has already been successfully performed by the rationale modification of toxins, thus generating crystals of new toxins with finely tuned properties, and that structural information, especially that obtained directly from crystals grown in vivo, is opening new routes of improvement. Moreover, recent works have highlight that *Bt* toxin crystals could entrap functional cargo proteins of high biotechnological and medical interest, calling for future investigations of additional crystalline toxin/cargo protein combinations to provide a comprehensive overview of the application boundaries of this system. Further studies are expected to explore the means to stabilize the cargo protein within different toxin-driven crystal lattices to tailor their function and to extend their use for structural biology, notably through X-ray crystallography.

## Figures and Tables

**Figure 1 toxins-13-00441-f001:**
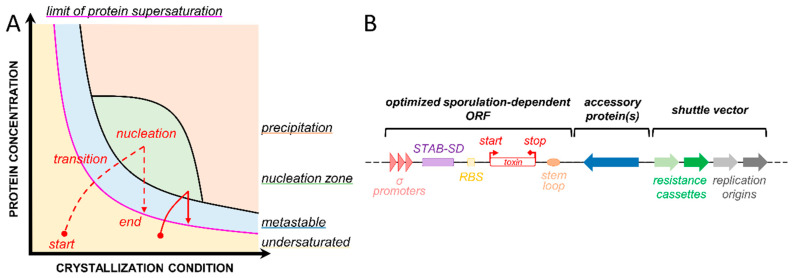
Crystallization requires a protein-specific combination of parameters. (**A**) The phase diagram represents the protein concentration as a function of the crystallization condition (concentration in precipitant, buffer concentration, etc.). This diagram is adapted from [10]. The concentration of the protein gradually increases to transition from the undersaturated zone (yellow) to the nucleation zone (green) for the crystal to start forming, thereby decreasing the concentration of protein in solution until the limit of protein supersaturation (purple) is reached and the crystal is stably formed (red lines in Figure 1A). Depending on the method used, the starting protein concentration, and the crystallization condition, one can obtain large single crystals (red dotted line) or multiple microcrystals (red solid line). (**B**) Example of a theoretical shuttle vector that can be used to produce a crystal of toxin in *Bt* [14,15]. It contains two origins of replication and two resistance cassettes that allow plasmid construction in one species (*Escherichia coli*) and toxin crystallization in *Bt*. The toxin gene expression is controlled by sporulation promoters, and the transcript is stabilized by a Shine–Dalgarno (SD)-like sequence from the 5′ UTR region of the *cry3aa* gene [16] and by a stem-loop in 3′ UTR [17]. Accessory proteins can be added to facilitate the folding and assembly and/or stabilize crystal contact interactions. More details on the role of these different features are provided in the first article of this back-to-back series and in the following references [18,19,20,21,22,23].

**Figure 2 toxins-13-00441-f002:**
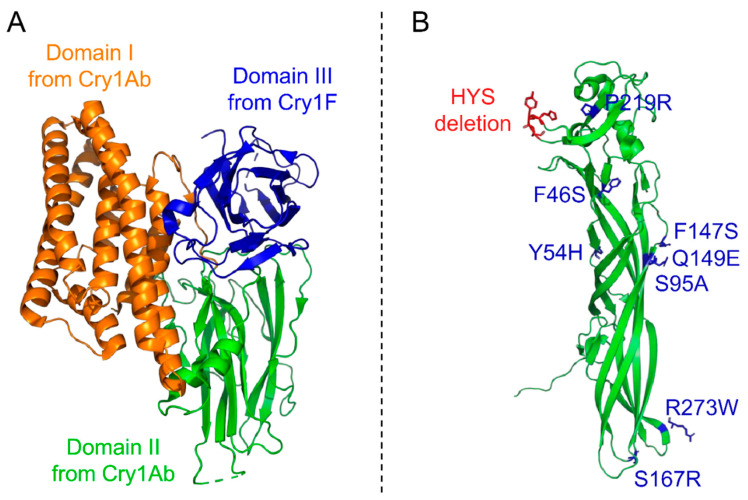
Structures of modified toxins with enhanced activity. (**A**) Structure of the tryptic core of the modified Cry1A.105 (PDB accession number: 6DJ4) toxin constituting domains I (orange) and II (green) from Cry1Ab and domain III (blue) from Cry1F [70]. (**B**) Structure of the Mpp51Aa2 toxin (formerly Cry51Aa2; PDB accession number: 5HD2) with eight mutated and three deleted residues to form the modified Cry51Aa2.834_16 toxin, highlighted in blue and green, respectively [78]. The 3D structures are represented in cartoon mode and the mutated residues in licorice mode using PyMOL Molecular Graphics System version 2.4.1.

**Figure 3 toxins-13-00441-f003:**
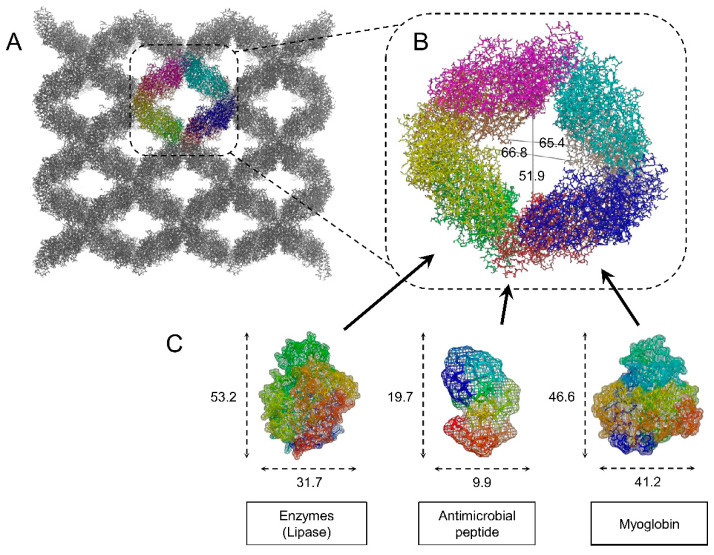
Cry3Aa crystal reveals large solvent channels accommodating proteins up to 50 Å in diameter. (**A**) Crystal packing of Cry3Aa monomers in vivo (PDB accession number: 4QX0). The eight monomers interacting to form the solvent channels are colored in red, blue, green, yellow, orange, cyan, magenta, and wheat tint, while all other monomers are in gray. (**B**) Zoomed view of the eight monomers with distances of the channel indicated in Å. (**C**) Structures of three proteins for which cocrystallization with Cry3Aa improved their stability, delivery, and/or activity: the catalytic enzyme *Proteus mirabilis* lipase (PML) [92], an antimicrobial peptide (dermaseptin) [95], and myoglobin [94] with the PDB accession numbers 4GW3, 2DD6, and 2SPL, respectively. The structures used here are illustrative only and not representative of the exact proteins used in the respective studies. Values indicate the length and width of the three proteins given in Å. The 3D structures from panels A and B are represented in licorice-sticks mode and those in panel C in both licorice-sticks and mesh modes using PyMOL Molecular Graphics System version 2.4.1.

**Figure 4 toxins-13-00441-f004:**
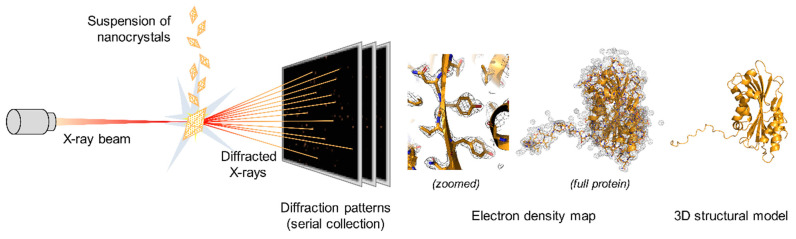
Principle of serial macromolecular X-ray crystallography. A suspension of homogenous crystals is injected serially in the course of an X-ray beam. Each crystal stands still and in random orientations during the exposure, enabling collection of only partial measurements of diffracted intensities. Partial intensities from thousands of indexed patterns from different microcrystals are then merged to generate a dataset that can be analyzed to generate the electron density map and allow determination of the three-dimensional structure of the protein. The 3D structures were generated using PyMOL Molecular Graphics System version 2.1.1 using the structure of Cyt1Aa protoxin with the PDB accession number 6T14 [88].

## Data Availability

Not applicable.

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
