# Peer review of "Can (We Make) Bacillus thuringiensis Crystallize More Than Its Toxins?"

_toxins, 2021, doi:10.3390/toxins13070441_

Round 1

Reviewer 1 Report

The ability of Bacillus thuringiensis (Bt) to produce numerous crystal toxins has significantly contributed to the development of biological insecticide formulations. Considering the importance of protein crystal structure and its impact in the scientific field, developing finely tuned and reliable crystallization will fulfill the gap or overcome the hurdles prevalent in the current in-vitro crystallization process. Here, the author explored the hijacking of the crystallization machinery of Bt to generate crystalline formulation proteins for three different purposes including the development of improved bioinsecticide crystal toxins, functionalize crystals with specific characteristics for biotechnological and medical applications and, generating microcrystals of custom proteins for structural biology.

The author has nicely presented the information in this review paper. I like how the author has provided directions for future investigations in promising but, not explored domains. Overall nice paper!

Author Response

Dear reviewer,

Thank you for your very positive feedback and for your support. I am glad you enjoyed this work.

Reviewer 2 Report

The review is interestingly and concisely written. To my knowledge, adequate references have been used to describe and summarise the field's current status, and sufficient background information is provided for the represented data. Additionally, the chosen outline is logical and easy to follow.

Minor comments (Grammar/ Typo etc.)

Line 86: "[..] Bt is to have evolved [...]"
Line 91: "[...] each of these toxins all leads to crystals [...]"
Line 132: "The same logic was used to produce in Bti the toxin Cry11B [...]"
Figure 3C is missing axes labelling.
Line 321: "[...] can resist to harsh treatments."
Line 323: "Such strain [...]"

Author Response

Dear reviewer,

Thank you for your feedback, I am very happy that you appreciated the article.

Below are the sentences modified accordingly:

Line 86 now reads: “Bt has evolved”

Line 91 now reads “each of these toxins leads to crystals”

Line 132: I struggle to modify this sentence without affecting its overall sense. I would welcome suggestion regarding the expected phrasing.

Figure 3C: The arrows are indicating distances in Angstrom. This information has been added in the title of the figure. It now reads: “Values are indicating the length and width of the three proteins, given in Å.”

Line 321 now reads: “can resist harsh treatments”

Line 323 now reads: “A strain with such characteristics”

This manuscript is a resubmission of an earlier submission. The following is a list of the peer review reports and author responses from that submission.

Round 1

Reviewer 1 Report

This manuscript provides a useful review of the knowledge on what is known about Bt crystal toxin protein regulation, expression, and crystallization. It is also generally well written. However, the premise of the title regarding whether Bacillus thuringiensis can crystallize more than its toxins is not fully met, and because of this the reader leaves somewhat disappointed at the end. This is partly because, as the review points out, there's not a lot of knowledge regarding the molecular mechanisms of how this crystallization takes place. More just what factors are involved, which doesn't provide insight into how to use the Bacillus thuringiensis system to produce crystals of proteins other than toxins. 

Given this, my suggestion is that the author reframe the review along a more traditional framework on the current knowledge of the crystallization of Bt crystal toxin proteins and their applications. He/she may end by discussing the work XFEL work on crystals in Bacillus, and suggest the possibility of using this to be extended to other proteins, but perhaps as a paragraph at the end of this section. 

Several other points.

(lines 294-295) For the legend and section "2.4. Crystallization in Bt depends on a fine equilibrium between these three factors" its actually unclear what factors the author means. Perhaps high expression, tendency to be crystallization prone, and the presence of accessory proteins, but it wasn't clear at this point in the reading. Fixing this may require editing the paragraph that introduces this material in lines 145-154, and restating what these factors are in line 296.

(lines 514-592) The section "Hijacking of the Bt crystallization machinery for the development of an in vivo microcrystallization platform" has a nice title, but the author provides no insight into how to do this - just how to analyze the in vivo crystals once they are formed. Perhaps this could be restructured into highlighting what has been done to solve structures of in vivo Cry protein crystals.

One general comment is that I found all of the figures to be of low quality, though this may be a consequence of the resolution of the copy used for the review. 

For Fig 1, the structures seems too small to garner any insight. Might mention that the 3 domain structure of many active Cry proteins is just a portion of the Cry1Ac structure.

For Fig 3A and B, the colors are too light so the associated text is difficult to read. Figures could be enlarged.

Author Response

This manuscript provides a useful review of the knowledge on what is known about Bt crystal toxin protein regulation, expression, and crystallization. It is also generally well written. However, the premise of the title regarding whether Bacillus thuringiensis can crystallize more than its toxins is not fully met, and because of this the reader leaves somewhat disappointed at the end. This is partly because, as the review points out, there's not a lot of knowledge regarding the molecular mechanisms of how this crystallization takes place. More just what factors are involved, which doesn't provide insight into how to use the Bacillus thuringiensis system to produce crystals of proteins other than toxins. 

Response. I thank the reviewer for acknowledging the importance of such work and for the quality of the writing. I totally understand the frustration of the reviewer by the lack of studies for some aspects of the mechanisms of crystallization and especially for the processes used to hijack the crystallization machinery. The aim of this article is to stimulate further research, which can start by frustration of lacking knowledge. This is why I think this article should rather be considered a review/opinion rather than a mere review because it goes beyond a comprehensive description of the published literature (part 2) as it also explores new aspects at the border of different fields that are underexplored (part 3) and proposes ways to further investigate them.

To comply with the reviewer’s comments, I added a paragraph (page 3 lines 103-118) describing in more details the purpose of each section in the global review/opinion piece. The latter term has also been added in different parts in the article (page 1 line 14, page 3 line 103 & 111, page 14, line 610). I believe that the conclusion is already clear in explaining which domains have been extensively explored and which ones deserve further investigation.

Given this, my suggestion is that the author reframe the review along a more traditional framework on the current knowledge of the crystallization of Bt crystal toxin proteins and their applications. He/she may end by discussing the work XFEL work on crystals in Bacillus, and suggest the possibility of using this to be extended to other proteins, but perhaps as a paragraph at the end of this section. 

Response. I understand the reviewer point of view but I respectfully disagree with it. The idea of this review/opinion arose from discussions with different scientists from different fields, and all questioned the potential of using Bt to crystallize other proteins. This review/opinion is therefore built to address – but not directly to answer – this question. The suggestion made by this reviewer would convert this review/opinion into yet another review on Bt crystallization mechanisms, which has already been done several times and more comprehensively (as highlighted page 3 lines 121-123). Here, the part 2 of the article developing these mechanisms lays the foundation for the part 3 of the article containing three major aspects that are not dissociable and of comparable importance.

Concerning the structural work using XFELs, it is involved in two distinct aspects that are logically presented and discussed in two different parts. The involvement of XFEL to solve toxins structures is greatly increasing our understanding of toxin crystallization process in Bt by solving structures directly from in vivo nanocrystals. This has been introduced several times in part 2 (pages 5-6, lines 197-256; page 8, lines 321-339) but the method is not described as this would have been misleading at this point. The XFEL approach is further described in part 3.3. Its history, functioning and relevance (page 12, lines 528-555) leaves room to the importance and limitations of in vivo crystallization (page 13, lines 565-582) to end by how Bt could be used for such purpose, with guidelines to do so (pages 13-14, lines 583-606). I honestly don’t think that I can develop more on both aspects, as it would become quite cumbersome.

Several other points.

(lines 294-295) For the legend and section "2.4. Crystallization in Bt depends on a fine equilibrium between these three factors" its actually unclear what factors the author means. Perhaps high expression, tendency to be crystallization prone, and the presence of accessory proteins, but it wasn't clear at this point in the reading. Fixing this may require editing the paragraph that introduces this material in lines 145-154, and restating what these factors are in line 296.

Response. I thank the reviewer for noticing this. Indeed, the title was a bit misleading. It has been changed to “Crystallization in Bt is a finely-tuned multifactorial phenomenon”. I modified the first sentence of the paragraph that now reads (pages 7-8, lines 307-309): “Each of the three factors presented above (i.e., the finely-regulated production of high quantities of toxins, their crystallization proneness and the facultative presence of accessory proteins) rarely solely drive toxins crystallization [52-54]”.

Concerning the paragraph page 4 lines 153-162, I erased the numbering of the last two factors presented as this did not correspond to the section numbering, which could have participated to the confusion. It also includes an additional last sentence (page 4 lines 162-165) for further increasing the clarity: “Hereafter, I develop in more details the diversity of the mechanisms used to produce a high quantity of toxins (section 2.1), the intrinsic crystalline characteristics of toxins (section 2.2), the role of accessory proteins (section 2.3) and eventually how a fine equilibrium between these three factors conditions toxins crystallization in Bt (section 2.4).”

(lines 514-592) The section "Hijacking of the Bt crystallization machinery for the development of an in vivo microcrystallization platform" has a nice title, but the author provides no insight into how to do this - just how to analyze the in vivo crystals once they are formed. Perhaps this could be restructured into highlighting what has been done to solve structures of in vivo Cry protein crystals.

Response. I understand the reviewer’s point of view. As described in response to above reviewer’s concerns, the importance of solving structures from in vivo grown crystals is yet another point (described in part 2 pages 5-6, lines 197-256; page 8, lines 321-339). The present paragraph aimed at presenting the reasons why getting crystals is important for solving protein structures and why Bt could meet these requirements (pages 577-584). Considering that no study has been published on this topic to date and as this article is built as a review/opinion, I present the arguments why Bt should be considered for custom crystal production in the future but I cannot develop more on the strategies to be used, which should capitalize on the strategies developed in the parts 3.1 and 3.2, as stated lines 581-584. This is why this section is presented at the end, the section 3 being built from the most studied and published (3.1) to the most promising yet unexplored field (3.3).

One general comment is that I found all of the figures to be of low quality, though this may be a consequence of the resolution of the copy used for the review. 

Response. I believe that the conversion into pdf has lowered the quality of the figures and impeded their proper visualization. They are provided separately in high quality to fix this problem before publication would this problem persists.

For Fig 1, the structures seems too small to garner any insight. Might mention that the 3 domain structure of many active Cry proteins is just a portion of the Cry1Ac structure.

Response. I am limited by the width of the page for the figure. With the highest quality, I did not notice problem for visualizing them even in printed format. I leave at the editor’s discretion the decision to enlarge the figures, considering that I provided the high-quality, which could easily accommodate size increases. Concerning the point on the toxins, this information is already provided in the text, when discussing the role of crystallization domains (part 2.2).

For Fig 3A and B, the colors are too light so the associated text is difficult to read. Figures could be enlarged.

Response. I apologize for this bad color choice. The associated text is now in black underlined with the corresponding color and the figures are enlarged as much as possible to fit the page. Like for Figure 1, the high-quality figure is provided separately would the editor want to add it as a larger figure in the text.

Reviewer 2 Report

The authors decribe in great detail the processes involved in the production of the crystal by Bt as their applications.

From the point of view of this reviewer, the review seems an introduction to a thesis or book chapter. The main idea of using Bt to encapsulate proteins of industrial or pharmacological interest (Section 3 manuscript) is just at the end of the manuscript and the take home message is diluted in too much information.

This reviewer believes that this review should be rewritten as opinion article focusing on section 3 of the present manuscript

Author Response

The authors decribe in great detail the processes involved in the production of the crystal by Bt as their applications.

From the point of view of this reviewer, the review seems an introduction to a thesis or book chapter. The main idea of using Bt to encapsulate proteins of industrial or pharmacological interest (Section 3 manuscript) is just at the end of the manuscript and the take home message is diluted in too much information.

This reviewer believes that this review should be rewritten as a minireview or opinion article focusing on section 3 of the present manuscript

Response. I thank the reviewer for acknowledging the effort to make this article a comprehensive piece of work containing all the information necessary to support its purpose. Indeed, the main purpose of this review/opinion is to address the section 3 of the manuscript. However, while I understand what the reviewer means, I respectfully disagree with the fact that the message is diluted. All the informations provided in the section 2 of this review are mandatory, providing the necessary knowledge foundation on which the section 3 of the manuscript is building upon. If we look at the numbers, the introduction is relatively short, with 1206 words (15% of total word count excluding the references section). The sections 2 and 3, respectively giving the necessary knowledge background and developing the innovative aspects of hijacking Bt crystallization machinery, are 3064 and 3691-words long, which corresponds to 38% and 45% of total word count. This means that the section 3, deemed the most important, is clearly already overrepresented. I believe that performing the modifications suggested by the reviewer would unbalance the article and would not permit readers that are not experts of the Bt system to understand its relevance for the questions addressed here. I however agree that the term review/opinion is particularly adapted to this article and it has been coined this term throughout the manuscript accordingly (page 1 line 14, page 3 line 103 & 111, page 14, line 610). An additional paragraph has been added at the end of the introduction to clarify the organization of the article and the relevance of each section (page 3, lines 103-118).

Round 2

Reviewer 1 Report

It seems that the author wishes to keep his/her general format - based on the identical title and abstract. Since my response was to change this since despite it being sexier, there was no evidence for this promise, it probably would be better for another reviewer to judge this in order to provide their different view. That reviewer might be more supportive which is fine.   Best,

Reviewer 2 Report

This reviewer understand perfectly, that the authors should not be agree with him. First of all this reviewer appreciate the work do it by the authors in trying to put on the table other applications of Bt.

This reviewer it is aware that in the research done by Bt research exist a bias to study the mode of action of Bt and not other possible applications of Bt outside of the agriculture. Since the main goal of this review is estimulate the discussion about other applications of Bt I belive that the format of the opinion article should provide the authors more freedom to develop their ideas.

Based on the review definition of the Toxins webpage journal ( Reviews: These provide concise and precise updates on the latest progress made in a given area of research. Systematic reviews should follow the PRISMA guidelines.), this reviewer believe that the current manuscript don't fit with the definition since the point 3 is quite speculative and constitute the core of the manuscript.

Finally this reviewer continue thinking that the manuscript is more and opinino article than a review.